

SciPost Phys. Lect. Notes 20 (2020)

# Entanglement spreading in non-equilibrium integrable systems

**Pasquale Calabrese**

SISSA and INFN, Via Bonomea 265, 34136 Trieste, Italy
International Centre for Theoretical Physics (ICTP), I-34151, Trieste, Italy

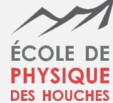

*Part of the Integrability in Atomic and Condensed Matter Physics*
*Session 111 of the Les Houches School, August 2018*
*published in the Les Houches Lecture Notes Series*

## Abstract

These are lecture notes for a short course given at the Les Houches Summer School on "Integrability in Atomic and Condensed Matter Physics", in summer 2018. Here, I pedagogically discuss recent advances in the study of the entanglement spreading during the non-equilibrium dynamics of isolated integrable quantum systems. I first introduce the idea that the stationary thermodynamic entropy is the entanglement accumulated during the non-equilibrium dynamics and then join such an idea with the quasiparticle picture for the entanglement spreading to provide quantitive predictions for the time evolution of the entanglement entropy in arbitrary integrable models, regardless of the interaction strength.

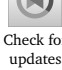
doi:10.21468/SciPostPhysLectNotes.20

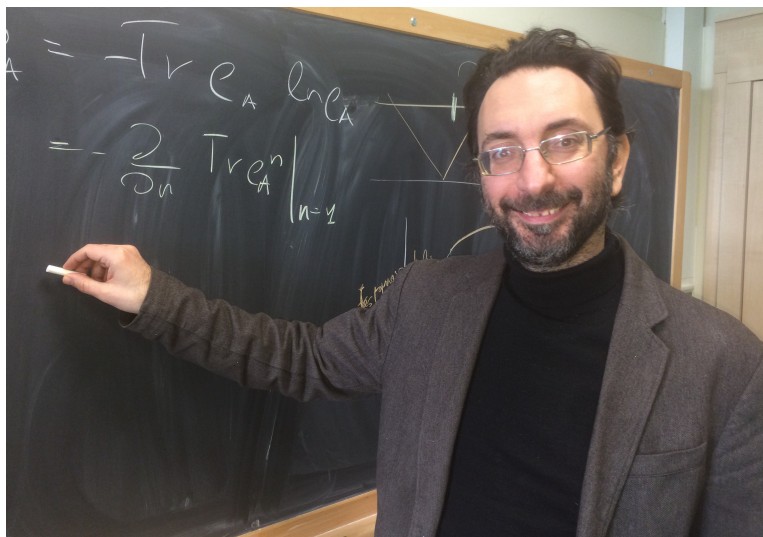

## 1 Introduction

Starting from the mid-noughties, the physics community witnessed an incredibly large theoretical and experimental activity aimed to understand the non-equilibrium dynamics of isolated many-body quantum systems. The most studied protocol is certainly that of a quantum quench [1,2] in which an extended quantum system evolves with a Hamiltonian $H$ after having being prepared at time $t = 0$ in a non-equilibrium state $|\Psi_0\rangle$, i.e. $[H, |\Psi_0\rangle\langle\Psi_0|] \neq 0$] ($|\Psi_0\rangle$ can also be thought as the ground state of another Hamiltonian $H_0$ and hence the name *quench*). At time $t$, the time evolved state is simply

$$|\Psi(t)\rangle = e^{-iHt}|\Psi_0\rangle, \tag{1}$$

where we work in units of $\hbar = 1$. A main question is whether for large times these many-body quantum systems can attain a stationary state and how this is compatible with the unitary time evolution of quantum mechanics. If a steady state is eventually reached (in some sense to be specified later), it is then natural to ask under what conditions the stationary properties are the same as in a statistical ensemble. This is the problem of thermalisation of an isolated quantum system, a research subject that has been initiated in 1929 by one of the fathers of quantum mechanics, John von Neumann, [3]. However, only in the last fifteen years the topic came to a new and active life, partially because of the pioneering experimental works with

cold atoms and ions which can probe closed quantum systems for time scales large enough to access the relaxation and thermalisation, see, e.g., the experiments in Refs. [4–13]. Nowadays, there are countless theoretical and experimental studies showing that for large times and in the thermodynamic limit, many observables relax to stationary values, as reported in some of the excellent reviews on the subject [14–19]. In some cases (to be better discussed in the following), these stationary values coincide with those in a thermal ensemble or suitable generalisations, despite the fact that the dynamics governing the evolution is unitary and the initial state is pure. Such relaxation is, at first, surprising because it creates a tension between the reversibility of the unitary dynamics and irreversibility of statistical mechanics.

In these lecture notes, I focus (in an introductory and elementary fashion) on the entanglement spreading after a quench. The interested reader can find excellent presentations of many other aspects of the problem in the aforementioned reviews [14–19]. Furthermore, I will not make any introduction to integrability techniques in and out of equilibrium because they are the subject of other lectures in the 2018 Les Houches school [20–23].

These lecture notes are organised as follows. In Sec. 2 it is shown how the reduced density matrix naturally encodes the concept of local relaxation to a stationary state. In Sec. 3 the entanglement entropy is defined and its role for the non-equilibrium dynamics is highlighted. In Sec. 4 we introduce the quasiparticle picture for the spreading of entanglement which is after applied to free fermionic systems (Sec. 5) and interacting integrable models (Sec. 6); in particular in Sec. 7 we briefly discuss some recent results within the entanglement dynamics of integrable systems.

## 2   Stationary state and reduced density matrix

The reduced density matrix is the main conceptual tool to understand how and in which sense for large times after the quench an isolated quantum system can be described by a mixed state such as the thermal one. Let us consider a non-equilibrium many-body quantum system (in arbitrary dimension). Since the time evolution is unitary, the entire system is in a pure state at any time (cf. $|\Psi(t)\rangle$ in Eq. (1)). Let us consider a spatial bipartition of the system into two complementary parts denoted as $A$ and $\bar{A}$. Denoting with $\rho(t) = |\Psi(t)\rangle\langle\Psi(t)|$ the density matrix of the entire system, the reduced density matrix is defined by tracing out the degrees of freedom in $\bar{A}$ as

$$\rho_A(t) = \mathrm{Tr}_{\bar{A}}\big[\rho(t)\big]. \tag{2}$$

The reduced density matrix $\rho_A(t)$ generically corresponds to a mixed state with non-zero entropy, even if $\rho(t)$ is a projector on a pure state. Its time dependent von Neumann entropy

$$S_A(t) = -\mathrm{Tr}[\rho_A(t)\log\rho_A(t)], \tag{3}$$

is called entanglement entropy and it is the main quantity of interest of these lectures. Some of its features will be discussed in the following section.

A crucial observation is that the physics of the subsystem $A$ is fully encoded in the reduced density matrix $\rho_A(t)$, in the sense that $\rho_A(t)$ is enough to determine all the correlation functions local within $A$. In fact, the expectation value of a product of local operators $\prod_i O(x_i)$ with $x_i \in A$ (which are the ones accessible in an experiment) is given by

$$\langle\Psi(t)|\prod_i O(x_i)|\Psi(t)\rangle = \mathrm{Tr}[\rho_A(t)O(x_i)]. \tag{4}$$

This line of thoughts naturally leads to the conclusion that the question "Can a close quantum system reach a stationary states?" should be reformulated as "Do local observables attain stationary values?".

Hence, the equilibration of a closed quantum system to a statistical ensemble starts from the concept of reduced density matrix. Indeed, we will say that, following a quantum quench, an isolated *infinite system* relaxes to a stationary state, if for *all finite* subsystems $A$, the limit of the reduced density matrix $\rho_A(t)$ for infinite time exists, i.e. if it exists

$$\lim_{t \to \infty} \rho_A(t) = \rho_A(\infty). \tag{5}$$

It is very important to stress that Eq. (5) implies a very precise order of limits; since the infinite time limit is taken for an infinite system, it means that the thermodynamic limit must be taken before the infinite time one; the two limits do not commute and phenomena like quantum recurrences and revivals prevent relaxation for finite systems (anyhow time-averaged quantities could still attain values described by a statistical ensemble). Another important observation is that although Eq. (5) is apparently written only for a subsystem $A$, it is actually a statement for the entire system. In fact, the subsystem $A$ is finite, but it is placed in an arbitrary position and it has an arbitrary (finite) dimension. Furthermore, the limit of a very large subsystem $A$ can also be taken, but only after the infinite time limit. Once again the two limits do not commute and their order is important. Summarising, there are three possible limits involved in the definition of the stationary state after a quantum quench; these limits do not commute and only one precise order leads to a consistent definition of equilibration of an isolated quantum system.

We are now ready to understand in which sense $\rho_A(\infty)$ may correspond to a statistical ensemble. A first guess would be that $\rho_A(\infty)$ is itself an ensemble density matrix (e.g. thermal). However, this definition would not be satisfactory because we should first properly consider boundary effects; moreover it would be valid only for thermodynamically large subsystems. We take here a different route following Refs. [24–28]. Let us consider a statistical ensemble with density matrix $\rho_E$ for the entire system. We can construct the reduced density matrix of a subsystem $A$ as

$$\rho_{A,E} = \mathrm{Tr}_{\bar{A}}(\rho_E). \tag{6}$$

We say that the stationary state is described by the statistical ensemble $\rho_E$ if, for any finite subsystem $A$, it holds

$$\rho_A(\infty) = \rho_{A,E}. \tag{7}$$

This implies that arbitrary local multi-point correlation functions within subsystem $A$, like those in Eq. (4), may be evaluated as averages with the density matrix $\rho_E$. This definition should not suggest that $\rho_E$ is the density matrix of the whole system that would be a nonsense because the former is a mixed state and the latter a pure one.

In these lectures, we are interested only into two statistical ensembles, namely the thermal (Gibbs) ensemble and the generalised Gibbs one. We say that a non-equilibrium quantum system thermalises after a quantum quench when $\rho_E$ is the Gibbs distribution

$$\rho_E = \frac{e^{-\beta H}}{Z}, \tag{8}$$

with $Z = \mathrm{Tr}\, e^{-\beta H}$. The inverse temperature $\beta = 1/T$ is not a free parameter: it is fixed by the conservation of energy. In fact, the initial and the stationary values of the Hamiltonian are equal, i.e.

$$\mathrm{Tr}[H\rho_E] = \langle \Psi_0 | H | \Psi_0 \rangle. \tag{9}$$

This equation can be solved for $\beta$, fixing the temperature in the stationary state. Once again, thermalisation leads to the remarkable consequence that all local observables will attain thermal expectations, but some non-local quantities will remain non-thermal for arbitrary large times. Generically, all non-integrable systems should relax to a thermal state, as supported

by theoretical arguments such as the eigenstate thermalisation hypothesis [29–32], by a large number of simulations (see, e.g., [33–48]), and by some cold atom experiments [4, 5, 9, 11]. However, there are some exceptional cases in which chaotic systems fail to thermalise like many-body localised ones [49, 50], or those in the presence of quantum scars [51–54], or when elementary excitations are confined [55–61].

The dynamics and the relaxation of integrable models are very different from chaotic ones because of the constraints imposed by the conservation laws. Integrable models have, by definition, an infinite number of integrals of motion in involution, i.e. $[I_n, I_m] = 0$ (usually one of the $I_m$ is the Hamiltonian). Consequently, rather than a thermal ensemble, the system for large time is expected to be described by a generalised Gibbs ensemble (GGE) [62] with density matrix

$$\rho_{\text{GGE}} = \frac{e^{-\sum_n \lambda_n I_n}}{Z}. \tag{10}$$

Here the operators $I_n$ form a complete set (in some sense to be specified) of integrals of motion and $Z$ is the normalisation constant $Z = \text{Tr}\, e^{-\sum_n \lambda_n I_n}$ ensuring $\text{Tr}\rho_{\text{GGE}} = 1$. As the inverse temperature for the Gibbs ensemble, the Lagrange multipliers $\{\lambda_n\}$ are not free, but are fixed by the conservation of $\{I_n\}$, i.e. they are determined by the (infinite) set of equations

$$\text{Tr}[I_n \rho_{\text{GGE}}] = \langle \Psi_0 | I_n | \Psi_0 \rangle. \tag{11}$$

In the above introduction to the GGE, we did not specify which conserved charges should enter in the GGE density matrix (10). One could be naively tempted to require that all linearly independent operators commuting with the Hamiltonian should be considered in the GGE, regardless of their structure; this is what one would do in a classical integrable system to fix the orbit in phase space. In this respect, the situation is rather different between classical and quantum mechanics. Indeed, any generic quantum model has too many integrals of motion, independently of its integrability. For example, all the projectors on the eigenstates $O_n = |E_n\rangle\langle E_n|$, are conserved quantities for all Hamiltonians since $H = \sum_n E_n |E_n\rangle\langle E_n|$. For a model with $N$ degrees of freedom, the number of these charges is exponentially large in $N$, instead of being linear, as one would expect from the classical analogue. All these integrals of motion cannot constrain the local dynamics and enter in the GGE, otherwise no system will ever thermalise and all quantum models would be, in some weird sense, integrable. The solution of this apparent paradox is that, as long as we are interested in the expectation values of *local* observables, only integrals of motion with some *locality* or *extensivity* properties must be included in the GGE [27, 28, 63, 64]. For examples, the energy and a conserved particle number must enter the GGE, while the projectors on the eigenstates should not. In the spirit of Noether theorem of quantum field theory, an integral of motion is local if it can be written as an integral (sum in the case of a lattice model) of a given local density. However, it has been recently shown that also a more complicated class of integrals of motion, known as quasilocal [65], have the right physical features to be included in the GGE [66,67]. The discussion of the structure of these new conserved charges is far beyond the goal of these lectures. Our main message here is that we nowadays have a very clear picture of which operators form a complete set to specify a well defined GGE in all integrable models, free and interacting.

We conclude this section by mentioning what happens for finite systems, also, but not only, to describe cold atomic experiments with only a few hundred constituents. When there is a maximum velocity of propagation of information $v_{\text{M}}$ (in a sense which will become clearer later), as long as we consider times such that $v_{\text{M}} t \lesssim L$, with $L$ the linear size of the system, all measurements would provide the same outcome as in an infinite system (away from the boundaries). Thus, a subsystem of linear size $\ell$ can show stationary values as long as $L$ is large enough to guarantee the existence of the time window $\ell \ll v_{\text{M}} t \lesssim L$.

# 3   Entanglement entropy in many-body quantum systems

In order to understand the connection between entanglement and the equilibration of isolated quantum systems, we should first briefly discuss the bipartite entanglement of many-body systems (see e.g. the reviews [68–71]). As we did in the previous section, let us consider an extended quantum system in a pure state $|\Psi\rangle$ and take a bipartition into two complementary parts $A$ and $\bar{A}$. Such spatial bipartition induces a bipartition of the Hilbert space as $\mathcal{H} = \mathcal{H}_A \otimes \mathcal{H}_{\bar{A}}$. We can understand the amount of entanglement shared between these two parts thanks to Schmidt decomposition. It states that for an arbitrary pure state $|\Psi\rangle$ and for an arbitrary bipartition, there exist two bases $|w_\alpha^A\rangle$ of $\mathcal{H}_A$ and $|w_\alpha^{\bar{A}}\rangle$ of $\mathcal{H}_{\bar{A}}$ such that $|\Psi\rangle$ can be written as

$$|\Psi\rangle = \sum_\alpha \lambda_\alpha |w_\alpha^A\rangle \otimes |w_\alpha^{\bar{A}}\rangle. \tag{12}$$

The Schmidt eigenvalues $\lambda_\alpha$ quantify the non-separability of the state, i.e. the entanglement. If there is only one non-vanishing $\lambda_\alpha = 1$, the state is separable, i.e. it is unentangled. Conversely, the entanglement gets larger when more $\lambda_\alpha$ are non-zero and get similar values.

Schmidt eigenvalues and eigenvectors allow us to write the reduced density matrix $\rho_A = \text{Tr}_{\bar{A}} |\Psi\rangle\langle\Psi|$ as

$$\rho_A = \sum_\alpha |\lambda_\alpha|^2 |w_\alpha^A\rangle\langle w_\alpha^A|, \tag{13}$$

and similarly for $\rho_{\bar{A}}$ with $|w_\alpha^{\bar{A}}\rangle$ replacing $|w_\alpha^A\rangle$. A proper measure of the entanglement between $A$ and $\bar{A}$ is the von Neumann entropy of $\rho_A$ or $\rho_{\bar{A}}$

$$S_A = -\text{Tr}\rho_A \log \rho_A = -\sum_\alpha |\lambda_\alpha|^2 \log |\lambda_\alpha|^2 = -\text{Tr}\rho_{\bar{A}} \log \rho_{\bar{A}} = S_{\bar{A}}, \tag{14}$$

which is known as *entanglement entropy* (hereafter log is the natural logarithm). Obviously many other functions of the Schmidt eigenvalues are proper measures of entanglement. For example, all the Rényi entropies

$$S_A^{(n)} \equiv \frac{1}{1-n} \log \text{Tr}\rho_A^n = \frac{1}{1-n} \log \sum_\alpha |\lambda_\alpha|^{2n}, \tag{15}$$

quantify the entanglement for any $n > 0$. These Rényi entropies have many important physical properties. First, the limit for $n \to 1$ provides the von Neumann entropy and, for this reason, they are the core of the replica trick for entanglement [72,73]. Then, for integer $n \geq 2$, they are the only quantities that are measurable in cold-atom and ion-trap experiments [11–13, 74–77] ($\text{Tr}\rho_A^2$ is usually referred as purity in quantum information literature). Finally their knowledge for arbitrary integer $n$ provides the entire spectrum of $\rho_A$ [78], known as entanglement spectrum [79].

Rigorously speaking entanglement and Rényi entropies are good entanglement measures in the sense that they are entanglement monotones [80]. While these lectures are not the right forum to explain what an entanglement monotone is (the interested reader can check, e.g., the aforementioned [80]), we want to grasp some physical intuition about the physical meaning of the entanglement entropy. To this aim, let us consider the following simple two-spin state

$$|\Psi\rangle = \cos(\alpha)|+-\rangle - \sin(\alpha)|-+\rangle, \tag{16}$$

with $\alpha \in [0, \pi/2]$. It is a product state for $\alpha = 0$ and $\alpha = \pi/2$ and we expect that the entanglement should increase with $\alpha$ up to a maximum at $\alpha = \pi/4$ (the singlet state). The reduced density matrix of one of the two 1/2 spins is

$$\rho_A = \cos^2(\alpha)|+\rangle\langle+| + \sin^2(\alpha)|-\rangle\langle-|, \tag{17}$$

with entanglement entropy

$$S_A = -\sin^2(\alpha)\log(\sin^2(\alpha)) - \cos^2(\alpha)\log(\cos^2(\alpha)), \tag{18}$$

which has all the expected properties and takes the maximum value $\log 2$ on the singlet state.

Let us now consider a many-body system formed by many spins $1/2$ on a lattice and a state which is a collection of singlets between different pairs of spins at arbitrary distances (incidentally these states have important physical applications in disordered systems [81]). All singlets within $A$ or $\bar{A}$ do not contribute to the entanglement entropy $S_A$. Each shared singlets instead counts for a $\log 2$ bit of entanglement. Hence, the total entanglement entropy is $S_A = n_{A:\bar{A}}\log 2$ with $n_{A:\bar{A}}$ being the number of singlets shared between the two parts. As a consequence, the entanglement entropy measures all these quantum correlations between spins that can be very far apart.

Let us now move back to non-equilibrium quantum systems and see what entanglement can teach us. The stationary value of the entanglement entropy $S_A(\infty) = -\mathrm{Tr}\rho_A(\infty)\log\rho_A(\infty)$ for a thermodynamically large subsystem $A$ is simply deduced from the reasoning in the previous section. Indeed, we have established that a system relaxes for large times to a statistical ensemble $\rho_E$ when, for any finite subsystem $A$, the reduced density matrix $\rho_{A,E}$ (cf. Eq. (6)) equals the infinite time limit $\rho_A(\infty)$ (cf. Eq. (5)). This implies that the stationary entanglement entropy must equal $S_{A,E} = -\mathrm{Tr}\rho_{A,E}\log\rho_{A,E}$. For a large subsystem with volume $V_A$, $S_{A,E}$ scales like $V_A$ because the entropy is an extensive thermodynamic quantity. Hence, $S_{A,E}$ equals the density of thermodynamic entropy $S_E = -\mathrm{Tr}\rho_E\log\rho_E$ times the volume of $A$. Given that $S_{A,E} = S_A(\infty)$, the stationary entanglement entropy has the same density as the thermodynamic entropy. In conclusion, we have just proved the following chain of identities

$$s \equiv \lim_{V\to\infty}\frac{S_E}{V} = \lim_{V_A\to\infty}\frac{\lim_{V\to\infty}S_{A,E}}{V_A} = \lim_{V_A\to\infty}\frac{\lim_{V\to\infty}S_A(\infty)}{V_A}. \tag{19}$$

From the identification of the asymptotic entanglement entropy with the thermodynamic one we infer that the non-zero *thermodynamic entropy of the statistical ensemble is the entanglement accumulated during the time* by any large subsystem. We stress that this equality is true only for the extensive leading term of the entropies, as in Eq. (19); subleading terms are generically different. The equality of the extensive parts of the two entropies has been verified analytically for non-interacting many-body systems [82–86] and numerically for some interacting cases [87–89].

## 4 The quasiparticle picture

In this section, we descibe the quasiparticle picture for the entanglement evolution [90] which, as we shall see, is a very powerful framework leading to analytic predictions for the time evolution of the entanglement entropy that are valid for an arbitrary integrable model (when complemented with a solution for the stationary state coming from integrability). This picture is expected to provide exact results in the space-time scaling limit in which $t, \ell \to \infty$, with the ratio $t/\ell$ fixed and finite.

Let us describe how the quasiparticle picture works [18,90]. The initial state $|\Psi_0\rangle$ has an extensive excess of energy compared to the ground state of the Hamiltonian $H$ governing the time evolution, i.e. it has an energy located in the middle of the many-body spectrum. The state $|\Psi_0\rangle$ can be written as a superposition of the eigenstates of $H$; for an integrable system these eigenstates are multiparticle excitations. Therefore we can interpret the initial state as a source of quasiparticle excitations. We assume that quasiparticles are produced in pairs of

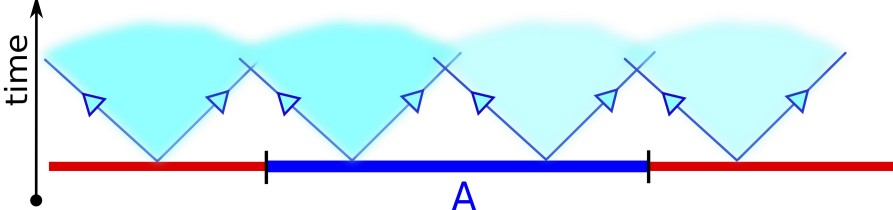

Figure 1: Quasiparticle picture for the spreading of entanglement. The initial state (at time $t = 0$) acts as a source of pairs of quasiparticles produced homogeneously throughout the system. After being produced, the quasiparticles separate ballistically moving with constant momentum-dependent velocity and spreading the entanglement.

opposite momenta. We will discuss when and why this assumption is correct for some explicit cases in the following, see Sec. 7.2 (clearly the distribution of the quasiparticles depends on the structure of the overlaps between the initial state and the eigenstates of the post-quench Hamiltonian). The essence of the picture is that particles emitted from different points are unentangled. Conversely, pairs of particles emitted from the same point are entangled and, as they move far apart, they are responsible for the spreading entanglement and correlations throughout the system (see Fig. 1 for an illustration). A particle of momentum $p$ has energy $\epsilon_p$ and velocity $v_p = d\epsilon_p/dp$. Once the two particles separate, they move ballistically through the system; we assume that there is no scattering between them and that they have an infinite lifetime (assumptions which are fully justified in integrable models [91]). Thus, a quasiparticle created at the point $x$ with momentum $p$ will be found at position $x' = x + v_p t$ at time $t$ while its entangled partner will be at $x'' = x - v_p t$.

The entanglement between $A$ and $\bar{A}$ at time $t$ is related to the pairs of quasiparticles that are shared between $A$ and $\bar{A}$ after being emitted together from an arbitrary point $x$. For fixed momentum $p$, this is proportional to the length of the interval (or region in more complicated cases) in $x$ such that $x' = x \pm v_p t \in A$ and $x'' = x \mp v_p t \in \bar{A}$. The proportionality constant depends on both the rate of production of pairs of quasiparticles of momentum $(p, -p)$ and their contribution to the entanglement entropy itself. The combined result of these two effects is a function $s(p)$ which depends on the momentum $p$ of each quasiparticle in the pair. This function $s(p)$ encodes all information about the initial state for the entanglement evolution.

Putting together the various pieces, the total entanglement entropy is [90]

$$S_A(t) \approx \int_{x' \in A} dx' \int_{x'' \in \bar{A}} dx'' \int_{-\infty}^{\infty} dx \int dp\, s(p) \delta\big(x' - x - v_p t\big) \delta\big(x'' - x + v_p t\big), \qquad (20)$$

which is valid for an arbitrary bipartition of the whole system in $A$ and $\bar{A}$. We can see in this formula all elements we have been discussing: (i) particles are emitted from arbitrary points $x$ (the integral runs over $[-\infty, \infty]$); (ii) they move ballistically as forced by the delta functions constraints over the linear trajectories; (iii) they are forced to arrive one in $A$ the other in $\bar{A}$ (the domain of integration in $x'$ and $x''$); (iv) finally, we sum over all allowed momenta $p$ (whose domain can depend on the model) with weight $s(p)$.

We specialise Eq. (20) to the case where $A$ is a single interval of length $\ell$. All the integrals over the positions $x, x', x''$ in Eq. (20) are easily performed, leading to the main result of the quasiparticle picture [90]

$$S_A(t) \approx 2t \int_{p>0} dp\, s(p) 2v_p \theta(\ell - 2v_p t) + 2\ell \int_{p>0} dp\, s(p)\theta(2v_p t - \ell)$$

$$= 2t \int_{2v_p t < \ell} dp\, s(p) 2v_p + 2\ell \int_{2v_p t > \ell} dp\, s(p). \quad (21)$$

Let us discuss the physical properties of this fundamental formula. For large time $t \to \infty$, the domain of the first integral shrinks to zero and so the integral vanishes (unless the integrand is strongly divergent too, but this is not physical). Consequently, the stationary value of the entanglement entropy is

$$S_A(\infty) \approx 2\ell \int_{p>0} dp\, s(p) = \ell \int dp\, s(p), \quad (22)$$

where in the rhs we used that $s(p) = s(-p)$ by construction. At this point, we assume that a maximum speed $v_M$ for the propagation of quasiparticles exists. The Lieb-Robinson bound [92] guarantees the existence of this velocity for lattice models with a finite dimensional local Hilbert space (such as spin chains). Also in relativistic field theories, the speed of light is a natural velocity bound. Since $|v(p)| \leq v_M$, the second integral in Eq. (21) is vanishing as long as $t < t^* = \ell/(2v_M)$ (the domain of integration again shrinks to zero). Hence, for $t < t^* = \ell/(2v_M)$ we have that $S_A(t)$ is *strictly linear* in $t$. For finite $t$ such that $t > t^*$, both integrals in Eq. (21) are non zero. The physical interpretation is that while the fastest quasiparticles (those with velocities close to $v_M$) reached a saturation value, slower quasiparticles continue arriving at any time so that the entanglement entropy slowly approaches the asymptotic value (22). The typical behaviour of the entanglement entropy resulting from Eq. (21) is the one reported in Fig. 2 where the various panels and curves correspond to the actual theoretical results for an interacting integrable spin chain (the anisotropic Heisenberg model, also known as the XXZ chain) that we will discuss in the forthcoming sections.

The last missing ingredients to make Eq. (21) quantitatively robust are the functions $s(p)$ and $v_p$ which should be fixed in terms of the quench parameters. The idea proposed in Ref. [93] (see also [94, 95]) is that $s(p)$ can be deduced from the thermodynamic entropy in the stationary state, using the fact that the stationary entanglement entropy has the same density as the thermodynamic one, cf. Eq. (19). To see how this idea works, we will apply it to free fermionic models in the next section and then to generic integrable models in the following one.

## 5 Quasiparticle picture for free fermionic models

The ab-initio calculation of entanglement entropy is an extremely challenging task. For Gaussian theories (i.e. non-interacting ones) it is possible to relate the entanglement entropy to the two-point correlation functions within the subsystem $A$ both for fermions and bosons [96–99]. Anyhow, for quench problems, extracting analytic asymptotic results from the correlation matrix technique is a daunting task that has been performed for some quenches in free fermions [82], but not yet for free bosons. We are going to see here that instead the quasiparticle picture provides exact analytic predictions in an elementary way, although not derived directly from first principles.

In this section, we consider an arbitrary model of free fermions. We focus on translational invariant models that can be diagonalised in momentum space $k$. It then exists a basis in which the Hamiltonian, apart from an unimportant additive constant, can be written as

$$H = \sum_k \epsilon_k b_k^\dagger b_k, \quad (23)$$

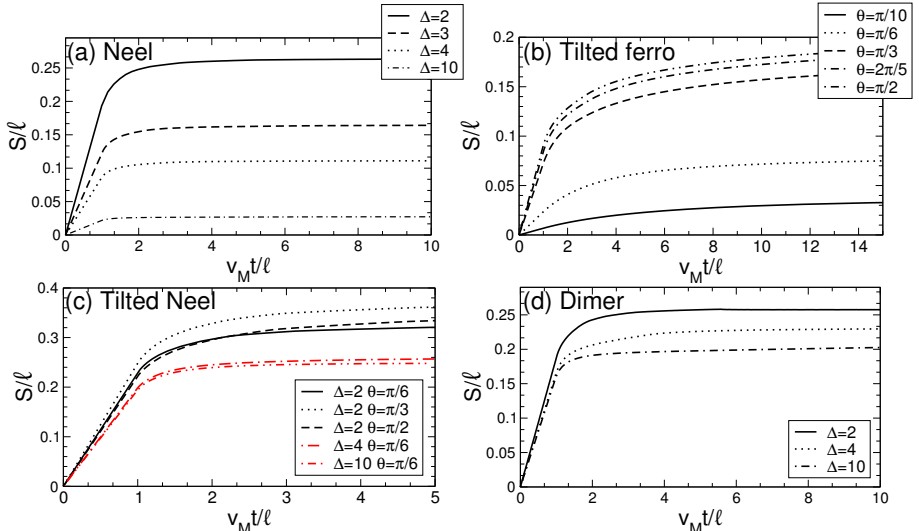

Figure 2: Quasiparticle prediction for the entanglement evolution after a global quench in the XXZ spin chain. In all panels the entanglement entropy density $S/\ell$ is plotted against the rescaled time $v_{\mathrm{M}}t/\ell$, with $\ell$ the size of $A$ and $v_{\mathrm{M}}$ the maximum velocity. Different panels correspond to different initial states, namely the Néel state (a), tilted ferromagnet with $Delta = 2$ (b), tilted Néel (c), and dimer state (d). Different curves correspond to different values of the chain anisotropy $\Delta > 1$ and tilting angles $\vartheta$ of the initial state. Figure taken from Ref. [94]

in terms of canonical creation $b_k^\dagger$ and annihilation $b_k$ operators (satisfying $\{b_k, b_{k'}^\dagger\} = \delta_{k,k'}$). The variables $\epsilon_k$ are single-particle energy levels.

We consider the quantum quench in which the system is prepared in an initial state $|\Psi_0\rangle$ and then is let evolve with the Hamiltonian $H$. For all these models, the GGE built with local conservation laws is equivalent to the one built with the mode occupation numbers $\hat{n}_k = b_k^\dagger b_k$ since they are linearly related [28]. Thus the local properties of the stationary state are captured by the GGE density matrix

$$\rho_{\mathrm{GGE}} \equiv \frac{e^{-\sum_k \lambda_k \hat{n}_k}}{Z}, \tag{24}$$

where $Z = \mathrm{Tr} e^{-\sum_k \lambda_k \hat{n}_k}$ (under some some reasonable assumptions on the initial state [26, 100, 101]).

The thermodynamic entropy of the GGE is obtained by elementary methods, leading, in the thermodynamic limit, to

$$S_{\mathrm{TD}} = L \int \frac{dk}{2\pi} H(n_k), \tag{25}$$

where $n_k \equiv \langle \hat{n}_k \rangle_{\mathrm{GGE}} = \mathrm{Tr}(\rho_{\mathrm{GGE}} \hat{n}_k)$ and the function $H$ is

$$H(n) = -n \log n - (1-n) \log(1-n). \tag{26}$$

The interpretation of Eq. (25) is obvious: $\rho_{GGE} = \bigotimes_k \rho_k$ with $\rho_k = \begin{pmatrix} n_k & 0 \\ 0 & 1-n_k \end{pmatrix}$, i.e. the mode $k$ is occupied with probability $n_k$ and empty with probability $1 - n_k$. Given that $\hat{n}_k$ is an integral of motion, one does not need to compute explicitly the GGE (24), but it is sufficient to calculate the expectation values of $\hat{n}_k$ in the initial state $\langle \psi_0 | \hat{n}_k | \psi_0 \rangle$ which equals, by construction, $n_k = \langle \hat{n}_k \rangle_{\mathrm{GGE}}$.

At this point, following the ideas of the previous sections (cf. Eq. (19)), we identify the stationary thermodynamic entropy with the density of entanglement entropy to be plugged in Eq. (21), obtaining the general result

$$S_A(t) = 2t \int\limits_{2|\epsilon_k'|t<\ell} \frac{dk}{2\pi} \epsilon_k' H(n_k) + \ell \int\limits_{2|\epsilon_k'|t>\ell} \frac{dk}{2\pi} H(n_k), \tag{27}$$

where $\epsilon_k' = d\epsilon_k/dk$ is the group velocity of the mode $k$. This formula is generically valid for arbitrary models of free fermions with the *crucial but rather general* assumption that the initial state can be written in terms of *pairs* of quasiparticles. More general and peculiar structures of initial states can be also considered, see Sec. 7.

Following the same logic, it is clear that Eq. (27) is also valid for free bosons (i.e. Hamiltonians like (23) with the ladder bosonic operators) with the minor replacement of the function $H(n)$ (26) with [94,95]

$$H_{\text{bos}}(n) = -n\log n + (1+n)\log(1+n). \tag{28}$$

## 5.1 The example of the transverse field Ising chain

Eq. (27) can be tested against available exact analytic results for the transverse field Ising chain with Hamiltonian

$$H = -\sum_{j=1}^{L}[\sigma_j^x \sigma_{j+1}^x + h\sigma_j^z], \tag{29}$$

where $\sigma_j^{x,z}$ are Pauli matrices and $h$ is the transverse magnetic field. The Hamiltonian (29) is diagonalised by a combination of Jordan-Wigner and Bogoliubov transformations [102], leading to Eq. (23) with the single-particle energies

$$\epsilon_k = 2\sqrt{1 + h^2 - 2h\cos k}. \tag{30}$$

We focus on a quench of the magnetic field in which the chain is initially prepared in the ground state of (29) with $h_0$ and then, at $t = 0$, the magnetic field is suddenly switched from $h_0$ to $h$. As in the general analysis above, the steady-state is determined by the fermionic occupation numbers $n_k$ given by [103]

$$n_k = \frac{1}{2}(1 - \cos\Delta_k), \tag{31}$$

where $\Delta_k$ is the difference of the pre- and post-quench Bogoliubov angles [103]

$$\Delta_k = \frac{4(1 + hh_0 - (h + h_0)\cos k)}{\epsilon_k \epsilon_k^0}, \tag{32}$$

with $\epsilon_k^0$ and $\epsilon_k$ the pre- and post-quench energy levels, respectively.

The quasiparticle prediction for the entanglement dynamics after the quench is then given by Eq. (27) with $n_k$ in Eq. (31). This result coincides with the *ab initio* derivation performed in [82]. The Ising model is only one of the many quenches in non-interacting theories of bosons and fermions in which the entanglement evolution is quantitatively captured by Eq. (27), as seen numerically in many cases [90, 104–114].

# 6 Quasiparticle picture for interacting integrable models

We are finally ready to extend the application of the quasiparticle picture to the entanglement entropy dynamics in interacting integrable models. We exploit the thermodynamic Bethe ansatz (TBA) solution of these models and remand for all the technicalities to other lectures in this school [20–22], or to the existing textbooks [115–118] on the subject. Here we just summarise the main ingredients we need and then move back to the entanglement dynamics.

## 6.1 Thermodynamic Bethe ansatz

In all Bethe ansatz integrable models, energy eigenstates are in one to one correspondence with a set of complex quasi momenta $\lambda_j$ (known as rapidities) which satisfy model dependent non-linear quantisation conditions known as Bethe equations. The solutions of the Bethe equations organise themselves into mutually disjoint patterns in the complex plane called *strings* [115]. Intuitively, an $n$-string solution corresponds to a bound state of $n$ elementary particles (i.e., those with $n = 1$). Each bound state (of $n$ particles) has its own quasi momentum $\lambda_\alpha^{(n)}$. The Bethe equations induce effective equations for the quantisation of the quasi momenta of the bound states known as Bethe-Takahashi equations [115]. In the thermodynamic limit, the solutions of these equations become dense on the real axis and hence can be described by smooth distribution functions $\rho_n^{(p)}(\lambda)$. One also needs to introduce the hole distribution functions $\rho_n^{(h)}(\lambda)$: they are a generalisation to the interacting case of the hole distributions of an ideal Fermi gas at finite temperature [115–118]. Because of the non-trivial (i.e. due to interactions) quantisation conditions, the hole distribution is not simply related to the particle one. Finally, it is also useful to introduce the total density $\rho_n^{(t)}(\lambda) \equiv \rho_n^{(p)}(\lambda) + \rho_n^{(h)}(\lambda)$.

In conclusion, in the thermodynamic limit a *macrostate* is identified with a set of densities $\boldsymbol{\rho} \equiv \{\rho_n^{(p)}(\lambda), \rho_n^{(h)}(\lambda)\}$. Each macrostate corresponds to an exponentially large number of microscopic eigenstates. The total number of equivalent microstates is $e^{S_{YY}}$, with $S_{YY}$ the thermodynamic Yang-Yang entropy of the macrostate [119]

$$S_{YY}[\boldsymbol{\rho}] \equiv L \sum_{n=1}^{\infty} \int d\lambda \Big[ \rho_n^{(t)}(\lambda) \ln \rho_n^{(t)}(\lambda) - \rho_n^{(p)}(\lambda) \ln \rho_n^{(p)}(\lambda) - \rho_n^{(h)}(\lambda) \ln \rho_n^{(h)}(\lambda) \Big]. \tag{33}$$

The Yang-Yang entropy is the thermodynamic entropy of a given macrostate, as it simply follows from a generalised microcanonical argument [119]. In particular, it has been shown that for in thermal equilibrium it coincides with the thermal entropy [115].

## 6.2 The GGE as a TBA macrostate

The generalised Gibbs ensemble describing the asymptotic long time limit of a system after a quench is one particular TBA macrostate and hence it is fully specified by its rapidities (particle and hole) distribution functions. There are (at least) three effective ways to calculate these distributions (see also the lectures by Fabian Essler [20]). The first one is based on the quench action approach [120,121], a recent framework that led to a very deep understanding and characterisation of the quench dynamics of interacting integrable models. This technique is based on the knowledge of the overlaps between the initial state and Bethe eigenstates. Starting from these, it provides a set of TBA integral equations for the rapidity distributions in the stationary state that can be easily solved numerically and, in a few instances, also analytically. In turns, the developing of such approach also motivated the determination of the exact overlaps in many Bethe ansatz solvable models [122–145]. Based on these overlaps, a lot of exact results for the stationary states have been systematically obtained in integrable models [122, 146–162]. We must mention that only thanks to the quench action solutions of

some quenches in the XXZ spin chain [150–153], it has been discovered that the GGE built with known (ultra)local charges [163–165] is insufficient to describe correctly [166, 167] the steady state; this result motivated and boosted the discovery of new families of quasi-local conservation laws that must be included in the GGE [66, 67, 168–170]. This finding is extremely important because when a complete set of charges is known, the stationary state can be built circumventing the knowledge of the overlaps required for quench action solution, as e.g. done in Refs. [171–179]. The direct construction of the GGE based on all the linear independent quasilocal conserved charges is the second technique to access the asymptotic TBA macrostate. The third technique is based on the quantum transfer matrix formalism [148, 149, 180, 181], but will not be further discussed here.

We finally mention that in the quench action formalism, the time evolution of local observables can be obtained as a sum of contributions coming from excitations over the stationary state [120]. This sum has been explicitly calculated for some non-interacting systems [120, 182, 183], but, until now, resisted all attempts for an exact computation in interacting models [146, 184] and hence it has only been numerically evaluated [185].

## 6.3 The entanglement evolution

As we have seen above, in interacting integrable models there are generically different species of quasiparticles corresponding to the bound states of $n$ elementary ones. According to the standard wisdom (based, e.g., on the $S$ matrix, see [91]), these bound states must be treated as independent quasiparticles. It is then natural to generalise Eq. (21), for the entanglement evolution with only one type of particles, to the independent sum of all of them, resulting in

$$S_A(t) = \sum_n \Big[ 2t \int_{2|v_n|t<\ell} d\lambda \, v_n(\lambda) s_n(\lambda) + \ell \int_{2|v_n|t>\ell} d\lambda \, s_n(\lambda) \Big], \tag{34}$$

where the sum is over the species of quasiparticles $n$, $v_n(\lambda)$ is their velocity, and $s_n(\lambda)$ the entropy density in rapidity space (the generalisation of $s(p)$ in Eq. (21)). To give predictive power to Eq. (34), we have to device a framework to determine $v_n(\lambda)$ and $s_n(\lambda)$ in the Bethe ansatz formalism.

The first ingredient to use is that in the stationary state the density of thermodynamic entropy (see Eq. (33)) equals that of the entanglement entropy in (34). Since this equality must hold for arbitrary root densities, we can identify $s_n(\lambda)$ with the density of Yang-Yang entropy for the particle $n$, i.e.

$$s_n(\lambda) = \rho_n^{(t)}(\lambda) \ln \rho_n^{(t)}(\lambda) - \rho_n^{(p)}(\lambda) \ln \rho_n^{(p)}(\lambda) - \rho_n^{(h)}(\lambda) \ln \rho_n^{(h)}(\lambda). \tag{35}$$

Moreover, the entangling quasiparticles in (34) can be identified with the excitations built on top of the stationary state. Their group velocities $v_n(\lambda)$ depend on the stationary state, because the interactions induce a state-dependent dressing of the excitations. These velocities $v_n(\lambda)$ can be calculated by Bethe ansatz techniques [186], but we do not discuss this problem here (see [94, 186] for all technical details).

Eq. (34) complemented by Eq. (35) and by the proper group velocities $v_n(\lambda)$ is the final quasiparticle prediction for the time evolution of the entanglement entropy in a generic integrable model. This prediction is not based on an ab-initio calculation and should be thought as an educated conjecture. It has been explicitly worked out using rapidity distributions of asymptotic macrostates for several models and initial states [93, 94, 160, 181, 187]. Some examples for the interacting XXZ spin chains, taken from [94], are shown in Fig. 2. The validity of this conjecture has been tested against numerical simulations (based on tensor network techniques) for a few interacting models. In particular, in Refs. [93, 94], the XXZ spin chain

for many different initial states and for various values of the interaction parameter $\Delta$ has been considered. The numerical data (after the extrapolation to the thermodynamic limit) are found to be in perfect agreement with the conjecture (34), providing a strong support for its correctness. In Ref. [188], the quasiparticle conjecture (34) has been tested for a spin-1 integrable spin chain, finding again a perfect match. This latter example is particularly relevant because it shows the correctness of Eq. (34) also for integrable models with a nested Bethe ansatz solution.

We conclude the section stressing that Eq. (34) represents a deep conceptual breakthrough because it provides in a single compact formula how the entanglement entropy becomes the thermodynamic entropy for an arbitrary integrable model.

# 7 Further developments

In this concluding subsection, we briefly go through several generalisations for the entanglement dynamics based on quasiparticle picture that have been derived starting from Eq. (34). Here, we do not aim to give an exhaustive treatment, but just to provide to the interested reader an idea of the new developments and some open problems.

## 7.1 Rényi entropies

A very interesting issue concerns the time evolution of the Rényi entropies defined in Eq. (15). These quantities are important for a twofold reason: on the one hand, they represent the core of the replica approach to the entanglement entropy itself [72, 73], on the other, they are the quantities that are directly measured in cold atom and ion trap experiments [11–13, 74–77].

For non-interacting systems, the generalisation of the formula for the quasiparticle picture is straightforward. Taking free fermions as example, the density of thermodynamic Rényi entropy in momentum space in terms of the mode occupation $n_k$ is just [82, 189]

$$s^{(\alpha)}(n_k) = \frac{1}{1-\alpha} \ln[n_k^\alpha + (1-n_k)^\alpha]. \tag{36}$$

Consequently, the time evolution of the Rényi entropy is just given by the same formula for von Neumann one, i.e. Eq. (27), in which $H(n_k)$ is replaced by $s^{(\alpha)}(n_k)$.

One would then naively expect that something similar works also for interacting integrable models. Unfortunately, this is not the case because it is still not known whether the Rényi analogue of the Yang-Yang entropy (33) exists. In Ref. [189] an alternative approach based on quench action has been taken to directly write the stationary Rényi entropy. First, in quench action approach, the $\alpha$-moment of $\rho_A$ may be written as the path integral [189]

$$\mathrm{Tr}\rho_A^\alpha = \int \mathcal{D}\boldsymbol{\rho}\, e^{-4\alpha\mathcal{E}[\boldsymbol{\rho}]+S_{YY}[\boldsymbol{\rho}]}, \tag{37}$$

where $\mathcal{E}[\boldsymbol{\rho}]$ stands for the thermodynamic limit of the logarithm of the overlaps, $S_{YY}[\boldsymbol{\rho}]$ is the Yang-Yang entropy, accounting for the total degeneracy of the macrostate, and the path integral is over all possible root densities $\boldsymbol{\rho}$ defining the macrostates. The most important aspect of Eq. (37) is that the Rényi index $\alpha$ appears in the exponential term and so it shifts the saddle point of the quench action. There is then a modified quench action

$$\mathcal{S}_Q^{(\alpha)}(\boldsymbol{\rho}) \equiv -4\alpha\mathcal{E}(\boldsymbol{\rho}) + S_{YY}(\boldsymbol{\rho}), \tag{38}$$

with saddle-point equation for $\boldsymbol{\rho}_\alpha^*$:

$$\left.\frac{\delta\mathcal{S}_Q^{(\alpha)}(\boldsymbol{\rho})}{\delta\boldsymbol{\rho}}\right|_{\boldsymbol{\rho}=\boldsymbol{\rho}_\alpha^*} = 0. \tag{39}$$

Finally, the stationary Rényi entropies are the saddle point expectation of this quench action

$$S_A^{(\alpha)} = \frac{\mathcal{S}_Q^{(\alpha)}(\boldsymbol{\rho}_\alpha^*)}{1-\alpha} = \frac{\mathcal{S}_Q^{(\alpha)}(\boldsymbol{\rho}_\alpha^*) - \alpha \mathcal{S}_Q^{(1)}(\boldsymbol{\rho}_1^*)}{1-\alpha}, \tag{40}$$

where in the rhs we used the property that $\mathcal{S}_Q^{(1)}(\boldsymbol{\rho}_1^*) = 0$, to rewrite $S_A^{(\alpha)}$ in a form that closely resembles the replica definition of the entanglement entropy [72,73].

Eq. (39) is a set of coupled equations for the root densities $\boldsymbol{\rho}_\alpha^*$ that can be solved, at least numerically, by standard methods. This analysis has been performed for several quenches in the XXZ spin chain [190,191] and the results have been compared with numerical simulations finding perfect agreement.

The main drawback of this approach is that the stationary Rényi entropy for $\alpha \neq 1$ is not written in terms of the root distribution of the stationary state $\boldsymbol{\rho}_1^*$ for local observables. Since the entangling quasiparticles are the excitations on top of $\boldsymbol{\rho}_1^*$, to apply the quasiparticle picture we should first rewrite the Rényi entropy in terms of $\boldsymbol{\rho}_1^*$. Unfortunately, it is still not know how to perform this step. We mention that an alternative promising route to bypass this problem is based on the branch point twist field approach [192,193]. The solution of this problem is also instrumental for the description of the symmetry resolved entanglement after a quantum quench [194].

## 7.2 Beyond the pair structure

A crucial assumption to arrive at Eq. (22) for the entanglement evolution is that quasiparticles are produced in uncorrelated pairs of opposite momenta. This assumption is justified by the structure of the overlaps between initial state and Hamiltonian eigenstates found for many quenches both in free [82, 103, 196–198] and interacting models [122, 129–132, 144, 195]. Indeed, it has been proposed that this pair structure in *interacting* integrable models is what makes the initial state compatible with integrability [195] and, in some sense, makes the quench itself integrable (see [195] for details). This no-go theorem does not apply to non-interacting theories and indeed, in free fermionic models, it is possible to engineer peculiar initial states such that quasiparticles are produced in multiplets [161, 162] or in pairs having non-trivial correlations [199, 200]. In all these cases, it is possible to adapt the quasiparticle picture to write exact formulas for the entanglement evolution, but the final results are rather cumbersome and so we remand the interested reader to the original references [161, 162, 199, 200].

## 7.3 Disjoint intervals: Mutual information and entanglement negativity

Let us now consider a tripartition $A_1 \cup A_2 \cup \bar{A}$ of a many-body system (with $A_1$ and $A_2$ two intervals of equal length $\ell$ and at distance $d$ and $\bar{A}$ the rest of the system). We are interested in correlations and entanglement between $A_1$ and $A_2$. A first measure of the total correlations is the mutual information

$$I_{A_1:A_2} \equiv S_{A_1} + S_{A_2} - S_{A_1 \cup A_2}, \tag{41}$$

with $S_{A_{1(2)}}$ and $S_{A_1 \cup A_2}$ being the entanglement entropies of $A_{1(2)}$ and $A_1 \cup A_2$, respectively. Using the quasiparticle picture and counting the quasiparticles that at time $t$ are shared between $A_1$ and $A_2$, it is straightforward to derive a prediction for the mutual information which reads [90, 93, 94]

$$I_{A_1:A_2} = \sum_n \int d\lambda \, s_n(\lambda) \Big[ -2 \max((d+2\ell)/2, v_n(\lambda)t)$$

$$+ \max(d/2, v_n(\lambda)t) + \max((d+4\ell)/2, v_n(\lambda)t) \Big], \tag{42}$$

where $s_n(\lambda)$ and $v_n(\lambda)$ have been already defined for the entanglement entropy. An interesting idea put forward in the literature is that one can use this formula to make spectroscopy of the particle content [94, 160]. In fact, since the typical velocities of different quasiparticles $n$ are rather different, Eq. (42) implies that the mutual information is formed by a train of peaks in time; these peaks become better and better resolved as $d$ grows compared to $\ell$ which is kept fixed.

The mutual information, however, is not a measure of entanglement between $A_1$ and $A_2$. An appropriate measure of entanglement is instead the *logarithmic negativity* $\mathcal{E}_{A_1:A_2}$ [201] defined as

$$\mathcal{E}_{A_1:A_2} \equiv \ln \mathrm{Tr}|\rho_{A_1 \cup A_2}^{T_2}|. \tag{43}$$

Here $\rho_A^{T_2}$ is the partial transpose of the reduced density matrix $\rho_A$. The time evolution of the negativity after a quench in an integrable model has been analysed in Refs. [202, 203]. To make a long story short, the quasiparticle prediction is the same as Eq. (42) but with $s_n(\lambda)$ replaced by another functional $\varepsilon(\lambda)$ of the root densities. This functional is related to the Rényi-1/2 entropy. Hence, as discussed in Sec. 7.1, we know it only for free theories. Exact predictions for free bosons and fermions have been explicitly constructed in Ref. [203] and tested against exact lattice calculations, finding perfect agreement.

## 7.4 Finite systems and revivals

How the quasiparticle picture generalise to a finite system of total length $L$? Starting from Eq. (21), it is clear that the only change is to impose the periodic trajectories of the quasiparticles which are $x_\pm = [(x \pm v_p t) \bmod L]$. Using these trajectories, the final result is easily worked out as [204–207]

$$S_\ell(t) = \int_{\left\{\frac{2v_k t}{L}\right\} < \frac{\ell}{L}} \frac{dk}{2\pi} s(k) L \left\{\frac{2v_k t}{L}\right\} + \ell \int_{\frac{\ell}{L} \leq \left\{\frac{2v_k t}{L}\right\} < 1 - \frac{\ell}{L}} \frac{dk}{2\pi} s(k)$$
$$+ \int_{1 - \frac{\ell}{L} \leq \left\{\frac{2v_k t}{L}\right\}} \frac{dk}{2\pi} s(k) L \left(1 - \left\{\frac{2v_k t}{L}\right\}\right), \quad (44)$$

where $\{x\}$ denotes the fractional part of $x$, e.g., $\{7.36\} = 0.36$. This form has been carefully tested for free systems [205] in which it is possible to handle very large sizes. For interacting models, tensor network simulations work well only for relatively small values of $L$, but still the agreement is satisfactory [205]. We must mention that Eq. (44) also applies to the dynamics of the thermofield double [204, 208], a state which is of great relevance also for the physics of black holes [209]. Finally, the structure of the revivals in minimal models of conformal field theories is also known [210].

## 7.5 Towards chaotic systems: scrambling and prethermalisation

What happens when integrability is broken? Can we say something about the time evolution of the entanglement entropy? It has already been found, especially in numerical simulations, that, in a large number of chaotic systems, the growth of the entanglement entropy is always linear followed by a saturation, see e.g. [41, 211–218]. This behaviour is the same as the one found in the quasiparticle picture, that, anyhow, cannot be the working principle here because the quasiparticles are unstable or do not exist at all. Recently, an explanation for this entanglement dynamics has arisen by studying random unitary circuits [219, 220], systems in which the dynamics is random in space and time with the only constraint being the locality of interactions. In this picture, the entanglement entropy is given by the surface of the minimal space-time membrane separating the two subsystems. It has been proposed that this picture

should describe, at least qualitatively, the entanglement spreading in generic non-integrable systems [221]. Random unitary circuits have been used to probe the entanglement dynamics in many different circumstances, providing a large number of new insightful results for chaotic models. Their discussion is however far beyond the scope of these lecture notes

Although the prediction for the entanglement entropy of a single interval in an infinite system is the same for both the quasiparticle and the minimal membrane pictures, the two rely on very different physical mechanisms and should provide different results for other entanglement related quantities. In fact, it has been found that the behaviour of the entanglement of disjoint regions [222–225] or that of one interval in finite volume [205,219,220,226] is qualitatively different. For maximally chaotic systems, the mutual information and the negativity of disjoint intervals are constantly zero and do not exhibit the peak from the quasiparticle picture seen in Eq. (42). The explanation of this behaviour is rather easy: in non-integrable models, the quasiparticles decay and scatter and they cannot spread the mutual entanglement far away. It has been then proposed that the decay of the peak of the mutual information and/or negativity with the separation is a measure of the scrambling of quantum information [222–225], as carefully tested numerically [225]. Remarkably, such a peak and its decay with the distance has been also observed in the analysis of the experimental ion-trap data related to the negativity [227]. Also in the case of a finite size system, the decay and the scattering of the quasiparticles prevent them to turn around the system; consequently the dip in the revival of the entanglement of a single interval predicted by Eq. (44) is washed out [226]. In full analogy with the mutual information, the disappearance of such dip is a quantitive measure of scrambling [205].

A natural question is now what happens to the entanglement dynamics when the integrability is broken only weakly. In this case, one would expect the two different mechanisms underlying the above picture to coexist until the metastable quasiparticles decay. This problem has been addressed in Ref. [228] finding that, for sufficiently small interactions, the entanglement entropy shows the typical prethermalization behaviour [229–234]: it first approaches a quasi-stationary plateau described by a deformed GGE and then, on a separate timescale, its starts drifting towards its thermal value. A modified quasiparticle picture provides an effective quantitative description of this behaviour: the contribution of each pair of quasiparticles to the entanglement becomes time-dependent and can be obtained by quantum Boltzmann equations [233,234], see for details [228].

## 7.6 Open systems

So far, we limited our attention to isolated quantum systems, but it is of great importance to understand when and how the quasiparticle picture can be generalised to systems that interact with their surrounding. In this respect, a main step forward has been taken in Ref. [235] (see also [236]), where it was shown that the quasiparticle picture can be adapted to the dynamic of some open quantum systems. In these systems, the spreading of entanglement is still governed by quasiparticles, but the environment introduces incoherent effects on top of it. For free fermions, this approach provided exact formulas for the evolution of the entanglement entropy and the mutual information which have been tested against ab-initio simulations.

## 7.7 Inhomogeneous systems and generalised hydrodynamics

The recently introduced generalised hydrodynamics [237,238] (see in particular the lectures by Ben Doyon in this volume [239]) is a new framework that empower us to handle spatially inhomogeneous initial states for arbitrary integrable models (generalising earlier works in the context of conformal field theory [240,241]). For what concerns the entanglement evolution, the attention in the literature focused on the case of the sudden junction of two leads [242–

246] (e.g., at different temperatures, chemical potentials, or just two different states on each side). One of the main results is that while the rate of exchange of entanglement entropy coincides with the thermodynamic one for free systems [244] (in analogy to homogenous cases), this is no longer the case for interacting integrable models [245]. Exact formulas, taking into account the inhomogeneities in space and time (and consequently the curved trajectories of the quasiparticles) can be explicitly written down both for free [244] and interacting [245] systems, but they are too cumbersome to be reported here. We finally stress that such an approach applies to states with locally non-zero Yang-Yang entropy, otherwise the growth of entanglement is sub-extensive and other techniques should be used [247, 248].

# Acknowledgments

I acknowledge Vincenzo Alba and John Cardy because most of the ideas presented here are based on our collaborations. I also thank Stefano Scopa for providing me hand-written notes of my lectures in Les Housches. I acknowledge support from ERC under Consolidator grant number 771536 (NEMO).

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
