# Peer review of "Entanglement spreading in non-equilibrium integrable systems"

_SciPost Physics Lecture Notes, doi:SciPost Phys. Lect. Notes 20 (2020)_

## Round 1 · Referee Report · Anonymous (Referee 1) · 2020-10-3

Strengths

1-An excellent account of entanglement spreading in integrable quantum systems out of equilibrium.
2-All the relevant literature is cited, particularly useful to the newcomers to the field.
3-Very pedagogical structure of the notes.

Weaknesses

1-Some chapters have unclear motivation.
2-Some statements may not be clear to non-experts.

Report

This lecture course is an excellent summary of a very timely topic of entanglement spreading in integrable quantum systems out of equilibrium. The first three chapters give an in-depth introduction leading up to the discussion of entanglement entropy in many-body quantum systems. The rest of the material is concerned with the quasiparticle picture, giving a short background and recounting the most important advancements in the field.

Overall, I find these notes very well written and I like how they are structured: introductory material is given an in-depth treatment, while more difficult concepts are summarised and properly cited. This gives students and newcomers to the field a strong background and a good overview of the current state of the field. However, I also feel that there is not enough motivation given at the beginning of some chapters, which may confuse the students significantly as to why are we interested in some of the derivations and problems. There are also multiple statements which may seem puzzling to the newcomers.

I recommend publishing this text in SciPost Physics Lecture Notes, but only after minor additions and corrections listed below are addressed by the author.

Requested changes

1) Abstract: I would strongly suggest mentioning the word "quasiparticle picture" in the abstract, as this topic takes most of the discussions in these lecture notes.

2) Page 2, "despite the fact that the dynamics governing the evolution is unitary and the initial state is pure": It would be useful to explain why is this surprising, i.e. that unitary evolution should lead to deterministic dynamics, where one can rewind time and read off the initial non-equilibrium state of the system. "Thermalisation" implies, however, that such rewinding should not be possible.

3) Page 3, "Let us consider a spatial bipartition of the system": More motivation is needed here, specifically why are we interested in bipartition and local operators. One can, for example, mention that in experiments we can usually only access local observables. This then naturally implies that the question "do we see thermalisation in quantum systems?" should be reformulated to "do local measurements see thermalisation?".

4) Page 5, "only integrals of motion with some locality or extensivity properties must be included in the GGE": At this point, it would be useful to give a simple example of what integrals of motion can enter GGE, as a direct comparison to the example of projectors on the eigenstates, which cannot enter GGE.

5) Page 6, Eq. 15: It would be useful to mention that the natural logarithm used in the definition of entropies is sometimes changed to $\log_2$, or even, $\log_{10}$. $\log_2$ is often used in the context of quantum information, where the natural unit of entropy is the maximum entropy of a spin-half singlet state.

6) Page 6: "Then, for integer n >= 2, they are the only quantities that are measurable in cold-atom and ion-trap experiments": I would suggest mentioning here that the n=2 Renyi entropy is directly related to purity, a quantity of interest for quantum information science.

7) Figure 2: For panel (b), please indicate what is the $\Delta$ used.

8) Page 9, "(the anisotropic Heisenberg model)": Since in the figures and later in the lecture notes, the name "XXZ chain" is used, I would say "(the anisotropic Heisenberg model, also known as the XXZ chain)".

9) Page 10, "The interpretation of Eq. (25) is obvious": For inexperienced students, it may not be so obvious, so it would be good to mention that the form of H(n) can be derived easily considering a 2x2 diagonal density matrix with n and (1-n) on the diagonal, corresponding to occupied and unoccupied modes.

10) Page 11, "Jordan-Wigner and Bogoliubov transformations": I would strongly suggest putting references here. Even better, one could add a reference to a step-by-step derivation from Eq. 29 to Eq. 30.

11) Page 12, "Intuitively, a n-string solution corresponds to a bound state of n elementary particles with n = 1.": I am puzzled by the clause "with n=1". Surely, the definition of a generic n-string solution should not include this clause?

12) Page 14: I would strongly suggest promoting the whole subchapter 6.4 to chapter 7, as it serves as a conclusion for these lecture notes and an outlook for the future.

13) Page 17, Chapters 6.4.5 and 6.4.6: Although the author briefly mentions the topic of random quantum circuits, it would be good to stress here that there has been extraordinary interest in these systems in the recent years. They have been used to probe entanglement dynamics in various scenarios, e.g. competition between unitary evolution and measurements, which can lead to extensive or sub-extensive entanglement behaviours, a topic which also ties into Ch. 6.4.6 on open systems and effects of the environment.

14) Minor spelling/interpunction: * "Neither theorem" -> "Noether's theorem" (page 5) * add a period to the end of the sentence "Here we just summarise the main ingredients we need and then move back to the entanglement dynamics" (page 12) * "a n-string solution" -> "an n-string solution" (page 12) * add a period to the end of the paragraph in chapter 6.4.6 (page 18)

15) Minor editing of equations, as it is sometimes difficult to read: * cos x -> cos(x), sin x -> sin(x) in Eqs. 16-18 * space after the variables of integration in Eqs. 20-22, 34, 37, 42

---

## Round 1 · Referee Report · Anonymous (Referee 2) · 2020-10-7

Strengths

1- Very pedagogical 2-Easy to read for newcomers 3- Avoids technical discussions 4-Quite complete coverage of key topics and related questions 5-Very good and balanced literature coverage as a source for further reading

Weaknesses

No weaknesses as far as I can see.

Report

This review provides and excellent and very pedagogical introduction into the theoretical description of entanglement spreading in nonequyilibrium integrable systems. The basic concepts - nonequilibrium dynamics in quantum quenches, thermalization in closed quantum systems, eigenstate thermalizatiin hypothesis, notion of integrability in quantum systems, von-Neumann
and Renyi entanglement entropies, entanglement dynamics in free systems and its calculation, as well as the picture of entanglement spreading via quasiparticle production and ballistic spreading are introduced. The review is kept nontechnical and refers the reader to specialized literature where necessary.

The review further makes contact to experimental efforts with quantum gases and ion traps, discusses perspectives on how to describe entanglement spreading in nonintegrable systems and open systems and closes with a brief mentiuoning of generalized hydrodynamics as powerful tool to deal with spatially inhomogeneous systems.

In my opinion, this will be a very useful and excellent review on the topic and I recommend its publication in essentially its present form. Some minor points and optional suggestions are listed below for the author's consideration.

Requested changes

1 -Page 2: von Neumann is misspelled: Von instead of von.

2- Page 4, 5 lines below Eq 7: double "the"

3- References to numerical tests of ETH are perhaps a bit selective, in particular, since no recent papers are cited. I cannot make a specific recommendation about what to cite, but there are many other people who contributed significantly (e.g., Lea Santos, Peter Prelovsek, Lev Vidmar, Robin Steinigeweg, Jochen Gemmer, and others). This is not central to the review's main topic, though.

4- Page 5: is the question of completeness of sets of conserved operators settled for typical integrable models, such as e.g., the spin-1/2 Heisenberg chain? I am not aware of a mathematical proof, in particular, given the newly discovered quasi-local charges. See the sentence at the end of the 1st paragraph.

5- Page 5: Is "Neither" perhaps a typo and should read "Noether"?

6- Page 8, 2nd paragraph: missing "of" in "pairs quasiparticles"

7- In Sec. 6.4.7, one could add a reference to Phys. Rev. B 90, 075144 (2014)

8- The author could (optionally) add an outlook onto open technical and conceptual questions.

---

## Round 1 · Referee Report · Anonymous (Referee 3) · 2020-12-10

Strengths

Well written
Pedagogic
Self-contained

Weaknesses

None.

Report

In "Entanglement spreading in non-equilibrium integrable systems", the author summarizes a set of lectures given at the 2018 Les Houches school on "Integrability in Atomic and Condensed Matter Physics". These lecture notes provide a brief overview of the how the evolution of entanglement can be understood in a system after a quantum quench.

The notes begin with a definition of what is meant by thermalization in a closed quantum system (important so as to establish the framework of the discussion) and then follow with a brief discussion of the notion of entanglement entropy. The notes then move into their central section, a discussion of the quasi-particle picture of entanglement spreading. This picture provides an intuitive semi-classical description to the growth of entanglement after a quantum quench. It envisions entanglement growing because a quench, which pumps finite energy density into a system, creates pairs of counter-propagating quasi-particles which fly apart after the quench, so spreading entanglement (and correlations) throughout the system. This simple picture, originally developed by the author with John Cardy in the context of quenches in conformal field theories (CFTs), has proven to be remarkably robust and applied to sundry cases beyond that of mere CFTs.

Having developed the quasi-particle picture, the author then goes on to show that it is not a mere cartoon but that it can be used to provide a quantitatively accurate computation of the growth of entanglement entropy post-quench. He ends these notes with a number of limitations on the immediate applicability of the quasi-particle picture including to the post-quench evolution of the Renyi entropies and to the non-equilibrium dynamics of non-integrable systems.

I believe these notes meet the criteria set out for SciPost Lecture notes inasmuch as they cover a topic of current interest to the community and they provide a clear, well-referenced, introduction to the concept of entanglement spreading.

Requested changes

My only comments are minor ones, mostly asking for typos or solecisms in the English to be fixed. Page numbers here are with reference to v2 on the arXiv.

i) Eqn. 4: There is a product \prod_i missing on the r.h.s. of Eqn. 4.

ii) pg 4: "that would be a nonsense" -> "that would be nonsensical"

iii) top of pg. 6: "get similar values" -> "approximately equal."

iv) pg 7: It might be worth stressing that oppositely counter-propagating quasiparticles results from a translationally invariant quench where the finite energy density state that results has zero momentum.

vi) pg 11: "It then exists ..." -> "There then exists ..."

vii) pg. 13: "In turns, the developing ..." -> "Correspondingly, the development ..."

viii) pg. 15: "Unfortunately, it is still not know" -> "Unfortunately, it is still not known"

ix) Perhaps somewhere in the vicinity of Section 7.5 a paragraph discussing the results of the author's own Ref. 51 might make sense. This is a nice result and can be readily be understood in terms of the quasi-particle picture.

  • validity: top
  • significance: top
  • originality: -
  • clarity: top
  • formatting: excellent
  • grammar: excellent

Author:  Pasquale Calabrese  on 2020-12-11  [id 1073]

(in reply to Report 3 on 2020-12-10)
Category:
remark

I thank the referee for reading my Lectures notes and for the very positive comments.
Unfortunately, I cannot fix the minor eight typos she/he spotted because the manuscript is already published.

Pasquale Calabrese

---

## Round 2 · Author Response

Dear Editor,

I thank the referees for the prompt and very positive reports. Both referees ask for some very minor adjustments that I took in consideration as detailed below.

Best Regards,

Pasquale Calabrese

Answers to referee 2

1 -Page 2: von Neumann is misspelled: Von instead of von.

A: Thanks, fixed.

2- Page 4, 5 lines below Eq 7: double "the"

A: Fixed

3- References to numerical tests of ETH are perhaps a bit selective, in particular, since no recent papers are cited. I cannot make a specific recommendation about what to cite, but there are many other people who contributed significantly (e.g., Lea Santos, Peter Prelovsek, Lev Vidmar, Robin Steinigeweg, Jochen Gemmer, and others). This is not central to the review's main topic, though.

A: I added the more recent references 38-44. This should be enough.

4- Page 5: is the question of completeness of sets of conserved operators settled for typical integrable models, such as e.g., the spin-1/2 Heisenberg chain? I am not aware of a mathematical proof, in particular, given the newly discovered quasi-local charges. See the sentence at the end of the 1st paragraph.

A: This is beyond the goal of these notes. The issue has been settled in Ref. [61] (and some other papers by Balazs Pozsgay) for model with a simple Bethe ansatz. For nested Bethe ansatz systems, to the best of my knowledge, there is no proof yet, but the completeness of local+quasilocal charges is likely true.

5- Page 5: Is "Neither" perhaps a typo and should read "Noether"?

A: Fixed

6- Page 8, 2nd paragraph: missing "of" in "pairs quasiparticles"

A: Fixed

7- In Sec. 6.4.7, one could add a reference to Phys. Rev. B 90, 075144 (2014)

A: Done

8- The author could (optionally) add an outlook onto open technical and conceptual questions.

A: Thank you for the suggestions. Few open questions are spread out in Section 6.4, now 7 (e.g. Renyi entropies, negativity, etc). I found that discussing even more open problems is beyond the scope of the lecture notes and would lead to a real review.

Answers to Referee 1

1) Abstract: I would strongly suggest mentioning the word "quasiparticle picture" in the abstract, as this topic takes most of the discussions in these lecture notes.

A: Thanks for the suggestion. Done.

2) Page 2, "despite the fact that the dynamics governing the evolution is unitary and the initial state is pure": It would be useful to explain why is this surprising, i.e. that unitary evolution should lead to deterministic dynamics, where one can rewind time and read off the initial non-equilibrium state of the system. "Thermalisation" implies, however, that such rewinding should not be possible.

I added the sentence "Such relaxation is, at first, surprising because it creates a tension between the reversibility of the unitary dynamics and irreversibility of statistical mechanics."

3) Page 3, "Let us consider a spatial bipartition of the system": More motivation is needed here, specifically why are we interested in bipartition and local operators. One can, for example, mention that in experiments we can usually only access local observables. This then naturally implies that the question "do we see thermalisation in quantum systems?" should be reformulated to "do local measurements see thermalisation?".

A: I liked the suggestion and added, after Eq. (4) the sentence: "This line of thoughts naturally leads to the conclusion that the question Can a close quantum system reach a stationary states?'' should be reformulated asDo local observables attain stationary values?''."

4) Page 5, "only integrals of motion with some locality or extensivity properties must be included in the GGE": At this point, it would be useful to give a simple example of what integrals of motion can enter GGE, as a direct comparison to the example of projectors on the eigenstates, which cannot enter GGE.

A: I added the sentence "For examples, the energy and a conserved particle number must enter the GGE, while the projectors on the eigenstates should not."

5) Page 6, Eq. 15: It would be useful to mention that the natural logarithm used in the definition of entropies is sometimes changed to log_2, or even, log_10. log_2 is often used in the context of quantum information, where the natural unit of entropy is the maximum entropy of a spin-half singlet state.

A: After Eq. (14) I added "(hereafter $\log$ is the natural logarithm)"

6) Page 6: "Then, for integer n >= 2, they are the only quantities that are measurable in cold-atom and ion-trap experiments": I would suggest mentioning here that the n=2 Renyi entropy is directly related to purity, a quantity of interest for quantum information science.

A: I added the sentence: "(${\rm Tr} \rho_A^2$ is usually referred as purity in quantum information literature)"

7) Figure 2: For panel (b), please indicate what is the \Delta used.

A: Added \Delta=2 in the caption

8) Page 9, "(the anisotropic Heisenberg model)": Since in the figures and later in the lecture notes, the name "XXZ chain" is used, I would say "(the anisotropic Heisenberg model, also known as the XXZ chain)".

A: Done

9) Page 10, "The interpretation of Eq. (25) is obvious": For inexperienced students, it may not be so obvious, so it would be good to mention that the form of H(n) can be derived easily considering a 2x2 diagonal density matrix with n and (1-n) on the diagonal, corresponding to occupied and unoccupied modes.

A: I added a longer explanation after Eq. (25).

10) Page 11, "Jordan-Wigner and Bogoliubov transformations": I would strongly suggest putting references here. Even better, one could add a reference to a step-by-step derivation from Eq. 29 to Eq. 30.

A: I added a reference to the book of Sachdev, where one can found all details of the transformations leading to Eq. (30).

11) Page 12, "Intuitively, a n-string solution corresponds to a bound state of n elementary particles with n = 1.": I am puzzled by the clause "with n=1". Surely, the definition of a generic n-string solution should not include this clause?

A: Here I just meant that the elementary particles are those with n=1, I rephrased to be clearer.

12) Page 14: I would strongly suggest promoting the whole subchapter 6.4 to chapter 7, as it serves as a conclusion for these lecture notes and an outlook for the future.

A: Thank you. Indeed, already in the first writing I was hesitating a lot if doing it or not. Now it is done.

13) Page 17, Chapters 6.4.5 and 6.4.6: Although the author briefly mentions the topic of random quantum circuits, it would be good to stress here that there has been extraordinary interest in these systems in the recent years. They have been used to probe entanglement dynamics in various scenarios, e.g. competition between unitary evolution and measurements, which can lead to extensive or sub-extensive entanglement behaviours, a topic which also ties into Ch. 6.4.6 on open systems and effects of the environment.

A: At the end of Sec 7.5, I added the sentence "Random unitary circuits have been used to probe the entanglement dynamics in many different circumstances, providing a large number of new insightful results for chaotic models. Their discussion is however far beyond the scope of these lecture notes".

14) Minor spelling/interpunction: * "Neither theorem" -> "Noether's theorem" (page 5) * add a period to the end of the sentence "Here we just summarise the main ingredients we need and then move back to the entanglement dynamics" (page 12) * "a n-string solution" -> "an n-string solution" (page 12) * add a period to the end of the paragraph in chapter 6.4.6 (page 18)

A: done

15) Minor editing of equations, as it is sometimes difficult to read: * cos x -> cos(x), sin x -> sin(x) in Eqs. 16-18 * space after the variables of integration in Eqs. 20-22, 34, 37, 42

A: Done

---

## Round 2 · List of Changes

The list of changes is fully given in the reply to referee

---

## Editorial Decision

published